# Characteristics of Homebirth in Hungary: A Retrospective Cohort Study

**DOI:** 10.3390/ijerph191610461

**Published:** 2022-08-22

**Authors:** Girma A. Wami, Viktória Prémusz, György M. Csákány, Kovács Kálmán, Viola Vértes, Péter Tamás

**Affiliations:** 1Doctoral School of Health Sciences, Faculty of Health Sciences, University of Pécs, Vörösmarty u. 4, H-7621 Pécs, Hungary; 2ELKH-PTE Human Reproduction Scientific Research Group, University of Pécs, Édesanyák u. 17, H-7624 Pécs, Hungary; 3Department of Obstetrics and Gynaecology, Jahn Ferenc Hospital, Kövesút 1, H-1204 Budapest, Hungary; 4Department of Obstetrics and Gynaecology, Faculty of Medicine, University of Pécs, Édesanyák u. 17, H-7624 Pécs, Hungary

**Keywords:** home childbirth, institutional childbirth, pregnancy complications, pregnancy outcome

## Abstract

Homebirth is legal and has been regulated by law in Hungary since 2012. Despite the obvious advantages of homebirth, it has not yet been broadly accepted, due to various opinions related to safety and risks associated with giving birth outside of a hospital. Our study aimed at exploring both real maternal and feto-neonatal characteristics associated with Hungarian homebirths. A total of 2997 cases were considered in support of our retrospective cohort study. In the examined period, there was a significant, continual rise in the number of homebirths by a rate of 0.22% on average per year. Aggregated maternal complications (primary uterine inertia, prolonged second stage labour, and third stage haemorrhage) were prevalent among homebirth cases (1.29% vs. 0.72%, *p* < 0.05) and were associated with an average of 11.77% rate of transfer to a health care institution. On the other hand, the rate of operative (vaginal or caesarean) delivery was 26.31% among institutionalized births. A slightly better Apgar score and relatively high rate (20%) of caesarean deliveries were correlated with institutionalized births (*p* < 0.05). However, the overall intervention rate was lower among homebirths (0.11% vs. 42.57%) than institutional birth cases (*p* < 0.001). Overall, homebirth is a reliable option for childbirth for healthy and low-risk mothers with uncomplicated pregnancies, which is reflected in the increasing number of deliveries at home in Hungary. Furthermore, utilizing the experiences of countries where homebirth is a long-established method may further improve the outcome of homebirths in Hungary.

## 1. Introduction

Homebirth for cases with normal pregnancy and managed by a licenced midwife is a safe option for healthy, low-risk women [1,2]. The safest place for a woman to give birth to her baby is believed to be at a functional health facility, with a professionally trained birth attendant. However, during the recent global crisis or due to other reasons, many women may end up giving birth at home [3]. Homebirth is an act of a woman giving birth to a child in ones’ own home [4]. During the recent pandemic, many women felt detached from sexual and reproductive health services, due to quarantine protocols. Pregnant women were unwilling to come to their local health facilities, due to disruptions in transportation associated with nationwide lockdown measures while others shunned the hospital or clinic due to increased fear of the spread of infections [5]. Others worried about giving birth surgically [6], and some reported health care providers’ reluctance to support birthing women’s needs that conflict with their moral beliefs [7]. Hence, there can be a trend for women who prefer to deliver their newborns at home [8].

It is not possible to predict the outcome of a pregnancy accurately. Problems during pregnancy may include physical and mental conditions that can range from mild and annoying discomforts to severe, sometimes life-threatening, illnesses that affect the life of a mother and her newborn. For instance, birthweight that reflects intra-uterine growth is an important determinant factor for perinatal morbidity and mortality [9,10] and, in recent years, has been shown to be a marker of postnatal health risks [9]. Although much is known regarding the clinical management of labour and childbirth, less attention is given to what clinical interventions make women feel safe, comfortable, and positive regarding the experience in the birth of their babies [9]. At the same time, the high ratio of operative deliveries is a real problem in many countries. Women need to receive health care before and during pregnancy to decrease the risk of pregnancy complications [10]. The prenatal care performed by county midwives and obstetricians is well organized in Hungary. Thus, the selection of patients eligible for homebirth is determined easily. However, while some problems can be caused by or can be made worse while being pregnant, many problems are mild and do not progress, but they can harm the mother and her newborn when they do. To cite a specific instance, birthweight, which reflects intra-uterine growth retardation, is an important determinant factor regarding perinatal morbidity and mortality [11,12] and, in recent years, is a marker in identifying postnatal health risks [13].

In Hungary, homebirth was neither legal nor illegal until 2012. In October 2010, police detained a Hungarian gynaecologist named Agnes Geréb. She was given a two-year prison sentence in February 2012 by the Budapest appeals court for two home birth instances. One entailed being there at the delivery of twins, one of whom died. In the second instance, she helped deliver a baby who eventually died six months later after suffering from serious problems. Later, Geréb and her lawyer asked for a new trial, citing fresh expert submissions. At that time, Dr. János Áder, the president of Hungary, granted Dr. Agnes Geréb’s request for mercy and overturned her jail term. She is a well-regarded independent obstetrician and gynaecologist whose life’s work was the freedom of birthing. More significantly, her efforts led to legislation that made homebirth legalized in Hungary [14,15]. Since then, strict regulations determine the conditions of homebirth for both pregnant women and participants of medical services. Healthy women, aged between 18 and 40 years, with normal, uncomplicated, single pregnancy, who are able to reach a hospital in 20 min in case a complication happens, are eligible for homebirth [16,17]. During the perinatal period, the pregnant mother enters into a contract with two persons (obstetrician or midwife and also their substitutes) who are licensed and corresponded to obstetrical specifications and are also skilful with adult resuscitation and the management of the neonate [18].

Outside of Hungary, several countries are considering the reintroduction of homebirths. This is based on claims of equal safety at lower intervention rates, compared to institutional births in which overtreatment may be present [19]. Furthermore, psycho-social advantages associated with homebirth for the mother are also beneficial [20] and the introduction of effective interprofessional collaboration should be driven for a safe home birth [21]. When the care for a patient is cohesive, the patient certainly benefits, as the health care team has worked together as a team to best address the pregnant women’s needs, wants, and values [22,23,24]. 

Homebirth is not yet widely accepted in Hungary [25], and the criminal stories covered by the media before 2012 most likely had an impact on the overall opinions of homebirth in Hungary [1,26]. Thus, our study attempts to explore the real maternal and feto-neonatal conditions and outcome characteristics regarding Hungarian homebirths. 

## 2. Materials and Methods

### 2.1. Design and Sample Size

This is a comparative retrospective cohort study. We sourced data regarding homebirths (n = 1792 from 2012 through 2020) from the Hungarian obstetric database (also called the ‘Tauffer database”) [27]. The ‘Tauffer database’ is managed and made available to researchers through the efforts of the National Institute for Quality and Organizational Development in Health Care and Medicine (reference 76/2004, decree regarding the determination, collection, and analysis of health-related unidentifiable data; Ministry of Health, Social and Family Affairs, Hungary) [16]. Institutional birth data (n = 1205) obtained from the Obstetrics and Gynaecology Department of the University of Pécs were matched according to Hungarian criteria of homebirth. All mothers included in this study were older than 18 years of age yet younger than 40 years old if it is their first pregnancy. Additional prerequisites included soon-to-be mothers being within 37–41 weeks of gestation, it being single pregnancy, the foetuses being in the cephalic position, and there being no prior history of any form of complication(s) during pregnancy. Those who planned to give birth at home must have access to a health facility that is equipped with obstetrical and neonatal services within 20 min of travel. These were our criteria and are representative of the basic permissive standards regarding homebirth, as stated in Hungarian laws about homebirth [16]. 

Data related to outcome conditions and complications during pregnancy were collected by International Classifications of Diseases, 10th revision (ICD-10 codes) [28]. 

### 2.2. Variables

Maternal-related variables included were maternal age, parity (primi- and multi-parous), gestational age, previous abortion (one or more), mode of conception (spontaneous or artificial), types of rupture of membrane (preterm, term, or artificial) and mode of birth (spontaneous or instrumental). 

Feto-neonatal variables included were the gender of the newborn, stillbirth, Apgar score measured five minutes following birth, early neonatal death at <168 h, birth weight, and birth weight percentiles. They were the independent variables, alongside which we have a dichotomous dependent variable (birthplaces, i.e., homebirth and institutional birth). Intervention was a health care variable.

### 2.3. Outcome Measures

Two primary outcomes were identified. First, “intervention” during birth, which is operationalized as undergoing intrapartum operative vaginal birth, or caesarean section and intrapartum artificial rupture of the membrane (AROM) is represented as ‘INTER2’. Secondly, perinatal mortality, in which it becomes operational, is a combination of stillbirths, intrapartum deaths, and early neonatal mortality until 168 h following delivery. The pooled outcome measures (Intervention * perinatal mortality) were used to estimate the risk ratio. The other secondary outcomes include maternal complications and outcome conditions. 

### 2.4. Case-Mix Adjustment 

Studies addressing the benefits and drawbacks of homebirth can be challenged due to their observational study design without case-mix adjustment regarding interventions and outcomes and the exclusion of women from the analysis, who, according to standardized birth guidelines, should have been referred to before birth. 

The case-mix was represented by the prevalence of the “Big4” conditions representative of an important risk mediator. These four conditions are known to precede 85% of perinatal mortality. These four-neonatal conditions are congenital abnormalities, intra-uterine growth restriction (small for gestational age), preterm, and low Apgar score (<7, measured 5 min following delivery). In a system highlighted with optimal risk, at least two big conditions (“Big2”), i.e., small for gestational age and low Apgar score are still present [20]. 

When comparing mortality rates, the “Big2” case-mix adjustment is used. However, when comparing intervention rates, the intervention precedes the outcome regarding a low Apgar score. Therefore, a low Apgar score should be excluded, and an analysis is compiled. Then pooled analyses (Intervention * mortality case-mix advanced model) were used to determine the risk ratio among groups.

### 2.5. Data Analysis

The data were analysed using IBM SPSS statistics version 26. The excel datasets were cleaned and de-identified before exporting to SPSS. Descriptive statistics were generated using frequencies, percentages, means, and standard deviations. Data of continuous parametric variables were presented as a mean ± standard deviation. The results of the chi-square were presented in APA format [29].

In consideration of statistical analysis, we used logistic regression models. *Model 1*, a binary logistic regression analysis, was presented as a crude odds ratio (COR), and *Model 2*, the multivariable logistic regression, was presented as an adjusted odds ratio (AOR) after adjusting for the confounders. In our study, a two-sided *p* value < 0.05 was considered statistically significant at a 95% confidence interval. All the explanatory variables with a threshold of *p* < 0.20 on a binary logistic regression model were fitted to a multivariable regression model and adjusted for confounders. Statistical significance was also cross-checked using backwards and forward stepwise regression analysis and demonstrated the same statistical significance.

Third, we compared the perinatal mortality rates after the “Big4” adjustment using an intention-to-treat-like approach. The intention-to-treat analysis is primarily used in RCTs [30]. However, we used the ‘intention-to-treat-like’ analysis approach, implying that all women having a home or institutional birth outcome were included, independent from later referral during labour. In consideration of this analysis, a nested multiple stepwise regression model (stepwise analysis; inclusion *p* < 0.20; exclusion *p* > 0.20) was used (model 1). Additionally, a pseudo-multicollinearity test was performed before running a multivariate logistic regression analysis and none were multicollinear. 

## 3. Results

### 3.1. Baseline Characteristics of Participants

A total of 2997 women were included in our study. During the considered period, 1792 mothers who experienced homebirths were compared with 1205 mothers who experienced an institutionalized birth. Our data have shown that the homebirths slowly increased over time by a rate of 0.22% per year on average (see Figure 1).

In terms of homebirths, the mean maternal age at first delivery was 33.16 ± 4.71, they were multiparous (66.50%), and the majority experienced a spontaneous mode of childbirth (94.01%); whereas, for institutional childbirths, 29.69 ± 5.44 was the mean maternal age at first delivery, 55.90% were primiparous, and 888 (73.69%) experienced spontaneous vaginal deliveries. The chi-square test of independence showed advanced-age mothers (≥35 years) were more likely to deliver at home, compared with younger-aged women (<35), *Χ*^2^(1, *n* = 2997) = 85.58, *p* < 0.001. Notably, mothers who had no prior history of previous spontaneous abortion were more likely to experience homebirths (*p* < 0.001), whereas nearly 332 (18.50%) women who had homebirths, and 11 (0.90%) who had institutional births used artificial means to conceive, *Χ*^2^(2, *n* = 2997) = 220.56, *p* < 0.001 (see Table 1). 

### 3.2. Feto-Neonatal Birth Characteristics and Outcome

Of the total 2997 newborns, 1537 (51.30%) were male, 1460 (48.70%) were female, (99.89%) were born alive, and 3 (0.11%) were reported as fatal cases (stillborn) during childbirth. The mean Apgar score at 5 min was 9.87 (±0.61) at home and 9.92 (±0.31) at institutions, respectively. Newborns from mothers who experienced homebirths had a slightly lower Apgar score at 5 min than when compared with institutional births *Χ^2^*(2, *n* = 2997) = 15.78, *p* < 0.001 (see Table 2). 

The mean birth weight was slightly higher in-homebirths (3556.87 ± 439.29), compared to institutional births (3433.16 ± 426.74), *Χ*^2^(2, *n* = 2997) = 22.34, *p* < 0.001, of which, the majority were appropriate and large for gestational age (59.01%, 39.50%), while few were small for gestational age (1.50%), and they were below 10th birth weight percentiles (see Figure 2). 

Newborns with relatively high birth weight were more likely to be born at home when compared with institutionalized deliveries, *Χ*^2^(2, *n* = 2997) = 22.34, *p* < 0.001 (see Table 2). 

Of the five early neonatal deaths (<168 h afterbirth), three (0.20%) were from home and two (0.17%) were among institutional births, and the reported stillbirths (0.17%) and intrapartum death at the institution (0.17%) were among transferred cases.

### 3.3. Maternal and Feto-Neonatal Birth Outcome Conditions and Complications

Of the total 2997 singleton births, 1183 (66.02%) homebirths and 884 (73.36%) institutional births were reported to have no obstetric complications. A relatively higher number of mothers who experienced institutional births had prolonged first-stage labour (3.32%), perineal laceration during birth (6.72%), or obstetric laceration of the cervix (2.91%) and 35 (2.91%) were anaemic (*p* < 0.05). Third-stage haemorrhage and delayed and secondary postpartum haemorrhage (1.40%, 0.84%) were prevalent maternal conditions reported from home births, respectively (see Table 3). A relatively higher number of foetal conditions, including foetal heart rate anomaly and meconium-stained amniotic fluid, at 22 (1.23%) were reported from homebirth cases (see Table 3). Overall, the women who had homebirths were 1.28% times the risk of complications, compared to women who had institutionalized births.

Regarding other determinants, the operative birth rate at the institution was 0.26 (26.31%) and about 0.05 (5.39%) were vacuum deliveries, while the institutional caesarean section rate was 0.21 (20.90%) (see Table 4). Only 24 (1.34%) women who experienced homebirths had an intention to operative birth. On average, the institutional transfer rate was 11.77%. 

The intervention rate was lower among homebirth cases (0.11%) compared with institutionalized births (42.57%) (*p* < 0.001). The crude intervention risk was significantly lower for women who experienced homebirths (COR 0.02, [95%CI 0.01–0.06, *p* < 0.001]) compared with women who experienced institutional births (Table 4, model 1). All maternal and neonatal risk factors (except the presence of a history of abortion, mode of birth and ROM) showed a significant difference. The adjusted intervention risk ratio demonstrates the birthplace indeed has a significant effect on the likelihood of intervention (AOR 0.02, [95%CI 0.01–0.05, *p* < 0.001]) (Table 4, model 2). Perinatal mortality was 11 (0.61%) among homebirths and 6 (0.49%) among institutionalized births, however, has not demonstrated any significant association with birthplace (see Table 4).

## 4. Discussion

According to our study, home births have been a more common occurrence in Hungary over the past ten years. As a result, the average homebirth rate in Hungary is 0.22%, which is considerably low when compared with the Netherlands (17%), New Zealand (3.5%), Australia (0.3%), or the United Kingdom (2.4%) [20,31,32,33]. Studies have indicated homebirth choice is controversial and enshrouded in debate. Generally, of issues related to risk and safety in a well-integrated health care system, homebirth is also deemed safe for healthy, low-risk women [1,2]. 

In Hungary, very little research has been published regarding homebirths for a multitude of reasons, specifically, the lack of funding and institutional support. However, beyond Hungary, studies indicate women, who planned homebirths, have experienced a very low risk of instrumental vaginal birth and caesarean section, therefore, a higher probability of spontaneous vaginal delivery [19,34,35,36,37]. Our study has also shown that the majority of low-risk women who gave birth at home have experienced a spontaneous mode of delivery (*p* < 0.001). A study originating from four Nordic countries has shown the majority of low-risk multiparous women who experienced spontaneous birth in their previous pregnancies were more likely to give birth at home [34]. Moreover, low-risk pregnancies attended by qualified midwives bring in positive results among both maternal and newborn health levels, including low rates of obstetric intervention [35,36,37].

Additionally, our study demonstrates how advanced age mothers (aged ≥ 35years) were more likely to experience homebirths than when compared with younger age mothers (*p* < 0.001), and our finding is consistent with the other studies, which aptly substantiated a general indication of the increased number of late childbearing age women aged 35 and above [38,39,40]. Today, it is becoming common for women in developed countries to delay their childbearing age. This phenomenon is due to multiple factors, yet effective birth control methods significantly contribute to postponing motherhood [41]. 

In our study, obstetrical complications related to mothers were prevalent among homebirths, while, relatively speaking, neonatal-related pathologic conditions and complications were more frequent among homebirth cases (*p* < 0.05). Other studies have shown that complications were more likely among planned homebirths [37,42]. 

We found that 1.5% of newborns from mothers who experienced homebirths had a relatively low Apgar score. Studies have shown that a newborn with a low Apgar score reflects a greater risk for obstetric and pregnancy-related complications [19,35,39,43]. Additionally, our findings are consistent with a study by Chandra et al. [44] regarding differences in maternal characteristics and pregnancy outcomes, which has shown a significant association between a low Apgar score of 5 min or poor pregnancy outcomes. 

Studies from countries with long-term experiences with homebirth confirm that deliveries at home with low-risk mothers with no previous history of obstetric complications show similar outcomes to institutional childbirths [19,45].

In our study, the hospital transfer rate was 11.77%. A systematic review of a large number of studies has shown that homebirth with such a transfer rate is considered reasonable, and an indication for the system is well integrated and able to support a pregnant mother’s choice regarding the place of birth [46]. Women who planned homebirth were considerably low risk throughout labour, and less likely to be transferred to an institution for advanced obstetric care, which typically results in a prolonged duration in second-stage labour [35]. This condition again is associated with the use of episiotomy, which may also be associated with perineal lacerations and intrapartum haemorrhage [37,39]. Similarly, our study also showed primary uterine inertia, prolonged second-stage labour, and third-stage haemorrhage, which were reported in women who experienced homebirths. The shorter duration of second-stage labour regarding institutional births may be due to higher intervention rates (episiotomy and option for operative delivery). Additionally, our study showed that “Big2” pregnancies at home exhibit a mortality disadvantage, suggesting comparatively lower intervention rates. The occurrence of overtreatment in the institution cannot be excluded. However, the benefit of fewer interventions among the homebirth group seems to be counterbalanced by substantially increased rates of complications. Our findings were partially inconsistent with a study from the Netherlands, in which increased larger sample sizes (n = 146,752) demonstrated that planned home births, attended by registered community midwives, confirm the lower risk of medical intervention, resulting in equal rates of complications [20]. In the Netherlands, the current homebirth rate is historically low, and it has never been this low before, with people’s experiences largely contributing to reaching this phenomenon. Other possible explanations were due to the increased chances for women who planned homebirth to switch their birthplace to an institution following a medical condition they recently experienced just before or during labour. Nonetheless, the safety and risks related to homebirth are not well expounded upon, and are very much a topic of debate, and published literature also substantiates regional variability [19,34,35,36,38,39,47].

### 4.1. Strength

As far as we are aware, this is the first study of its sort to describe homebirth characteristics in Hungary.

Notably, case-mix adjustment and intention-to-treat approach resulted in the most important aspect and strengthened our study. Without adjusting for this, one risks confounding the issue by indication bias. 

### 4.2. Limitations

Tauffer database is a compulsory database, however, some outcome variables were missed (like estimated volume of blood loss and birth outcomes of transferred cases) and less likely to be compared.The NICU admission, maternal weight (BMI), reason(s) used to transfer cases, and one-minute Apgar scores were not recorded in the compulsory database regarding homebirth cases. Lack of detailed information regarding maternal dropout and transfer for obstetric care, midwifery experiences, training, and their practices implemented in monitoring and evaluating feto–maternal conditions before and during birth. Despite baseline matching the potential confounders and restriction to low-risk women in our study, the possibility of residual confounding cannot be excluded given an observational study.

## 5. Conclusions

In careful consideration of our findings, both maternal and foetal–neonatal outcome conditions were relatively better among institutional cases when considering a comparatively lower perinatal mortality rate and fewer maternal complications. However, these slightly better results were associated with a high intervention rate. However, further research may be needed if this difference is being observed due to less detection of risk groups. 

Midwives should be regularly trained regarding strict clinical guidelines to precisely identify danger signs of imminent complications and upon those conditions pursue immediate hospital transfer to successfully avoid avoidable complications. More detailed statistical evidence will probably promote an exploration of the way to further improve the homebirth conditions in Hungary. Moreover, considering the experiences of countries with long-lasting practices of homebirth would support one in reaching the highest level of this significant human event at home.

## Figures and Tables

**Figure 1 ijerph-19-10461-f001:**
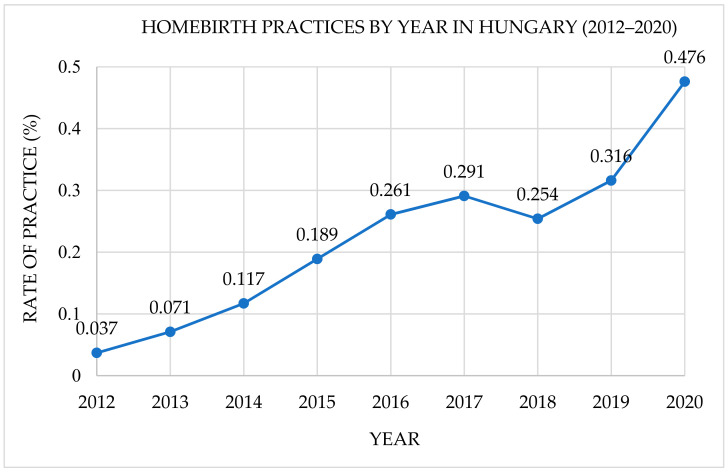
Trends of homebirth practices by year in Hungary (2012–2020).

**Figure 2 ijerph-19-10461-f002:**
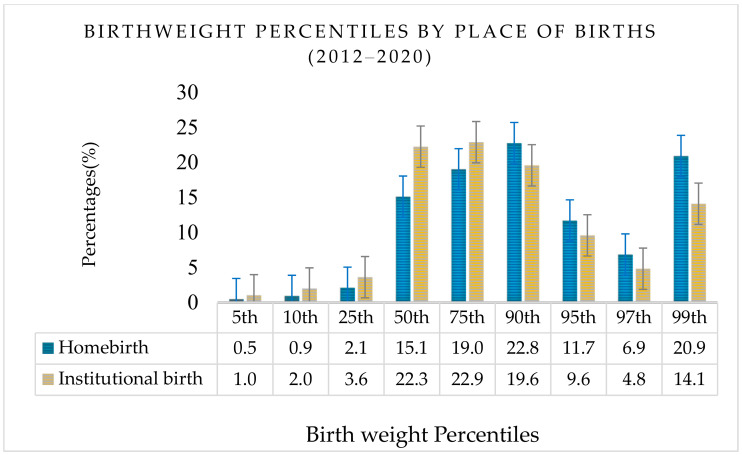
Birth weight percentiles by place of birth in Hungary (2012–2020).

**Table 1 ijerph-19-10461-t001:** Baseline characteristics of women who experienced home births and institutional births in Hungary from 2012–2020.

	Place of Birth		*Χ*^2^-Test of Independence (*n* = 2997)
	Home (*n* = 1792)	Institution (*n* = 1205)	
Maternal age (years)	33.16 ± 4.71 *	29.69 ± 5.44 *	*Χ*^2^(1) = 85.58 *p* < 0.001 φ = 0.17
Younger age (18–34.99)	1154 (64.40%)	965 (80.10%)	
Advanced age (≥35)	638 (35.60%)	240 (19.90%)	
Parity			*Χ*^2^(1) = 147.84 *p* < 0.001 φ = 0.22
Primi-parous	601 (33.50%)	674 (55.90%)	
Multiparous	1191 (66.50%)	531 (44.10%)	
History of spontaneous abortion			*Χ*^2^(2) = 18.67 *p* < 0.001 φ = 0.08
No	1561 (87.10%)	980 (81.30%)	
One-time	186 (10.41%)	181 (15.01%)	
Recurrent (≥2×)	45 (2.50%)	44 (3.70%)	
Mode of conception			*Χ*^2^(1) = 220.56 *p* < 0.001 φ = 0.27
Spontaneous	1460 (81.50%)	1194 (99.10%)	
Artificial	332 (18.50%)	11 (0.90%)	

(*) the result is presented as a mean SD, *Χ*^2^(df)-Pearson-chi square (degree of freedom), and φ-is value of Cramer’s V (indicating measures of association). φ = 0 depicts no association.

**Table 2 ijerph-19-10461-t002:** Characteristics and outcomes of the newborns from mothers who experienced home birth and institutional births in Hungary, 2012–2020.

	Place of Birth		*Χ*^2^-Test of Independence (*n* = 2997)
	Home (*n* = 1792)	Institution (*n* = 1205)	
Gender of the new-born			*Χ*^2^(1) = 0.68 *p* = 0.411 φ = 0.02
Male	908 (50.70%)	629 (52.30%)	
Female	884 (49.29%)	576 (47.79%)	
Apgar score (5 min)	9.87(±0.61)	9.92 (±0.31)	*Χ*^2^(2) = 15.78 *p* < 0.05 φ = 0.07
low < 7 score	27 (1.50%)	1 (0.10%)	
Normal ≥ 7 score	1746 (97.43%)	1200 (99.60%)	
Missed value	19 (1.10%)	4 (0.30%)	
Stillbirth			
No	1783 (99.50%)	1203 (99.83%)	-
Yes	9 (0.50%)	2 (0.17%)	
Intrapartum death			
No	1791 (99.94%)	1203 (99.83%)	-
Yes	1 (0.06%)	2 (0.17%)	
Birth weight (grams)	3556.87 ± 439.29	3433.16 ± 426.74	*Χ*^2^(2) = 22.34 *p* < 0.001 φ = 0.09
Low birth weight	6 (0.30%)	11 (0.90%)	
Average birth weight	1503(83.90%)	1070 (88.80%)	
High birth weight	283 (15.80%)	124 (10.30%)	
Birth weight percentile			*Χ*^2^(2) = 40.34 *p* < 0.001 φ = 0.12
SGA	26 (1.50%)	37 (3.10%)	
AGA	1058 (59.01%)	818 (67.90%)	
LGA	708 (39.50%)	350 (29.01%)	
Early neonatal death < 168 h			
No	1789 (99.80%)	1203 (99.83%)	-
Yes	3 (0.20%)	2 (0.17%)	

(-) indicates unsuitable regarding the chi-square model, and the expected count less than five is >20%, *Χ*^2^(df)-Pearson-chi square (degree of freedom), φ-is ‘phi- or Cramer’s V’-indicating measures of association (φ = 0 shows no association). Acronym and abbreviations: AGA: Appropriate for gestational age (weight between 10th–90th percentiles), LGA: Large for gestational age (≥90th percentiles) and SGA: Small for gestational age (<10th percentiles).

**Table 3 ijerph-19-10461-t003:** Comparisons of maternal and feto-neonatal homebirth and institutional birth-related conditions and complications in Hungary, 2012–2020.

Birth-Related Conditions and Complications (ICD-10)	Place of Birth	
	Home (*n* = 1792)	Institution (*n* = 1205)
No-obstetric complications **	1183 (66.02%)	884 (73.36%)
Primary uterine inertia	18 (1.01%))	5 (0.41%)
Prolonged first stage labour **	10 (0.56%)	40 (3.32%)
Prolonged second stage of labour *	26 (1.45%)	20 (1.66%)
Obstructed labour due to incomplete rotation of the fetal head	2 (0.11%	5 (0.41%)
Obstructed labour due to shoulder dystocia	3 (0.17%)	8 (0.66%)
Obstructed labour due to feto-pelvic disproportion	3 (0.17%)	7(0.58%)
Intrapartum haemorrhage (other)	5 (0.28%)	2 (0.17%)
First-degree perineal laceration during birth **	59 (3.29%)	81 (6.72%)
Second-degree perineal laceration during birth *	9 (0.50%)	14 (1.16%)
Third-degree perineal laceration during birth	9 (0.50%)	7 (0.58%)
Perineal laceration during birth (Unspecified)	8 (0.47%)	6(0.50%)
Obstetric laceration of the cervix	5 (0.28%)	35(2.91%)
Third stage haemorrhage **	25 (1.40%)	12 (0.10%)
Delayed and secondary postpartum haemorrhage	15 (0.84%)	5 (0.41%)
Retained placenta without haemorrhage	3 (0.18%)	7 (0.58%)
Anaemia *	10 (0.56%)	35 (2.91%)
FHR anomaly and meconium in the amniotic fluid	22 (1.23%)	13 (1.08%)

ICD-10 codes with frequency (*n* < 5) both at home and institution were not reported. FHR—Foetal heart rate: * *p* value < 0.05; ** *p* value < 0.01.

**Table 4 ijerph-19-10461-t004:** Summary statistics of women and feto-neonatal home and institutional birth characteristics and outcomes: Pooled risk measures (Intervention (C/s and AROM) * Perinatal mortality); using intention-to-treat-like approach and case-mix adjustment.

	Total (*n* = 2997)	Intervention *n* (%)	Mortality *n* (%)	*Model 1*		*Model 2* (R^2^ = 0.876)		
				COR (95%CI)	*p*	*β*	AOR (95%CI)	*p*
Place of birth		***	0.679					
Home	1792	2 (0.11%)	11(0.61%)	0.02 (0.01–0.06)	***	−6.95	0.02 (0.01–0.05)	***
Institution (Ref)	1205	513(42.57%)	6 (0.49%)	1			1	
Maternal age (years)		***	-					
Young age (18–34.99) (Ref)	2119	420 (19.80%)	11 (0.51%)	1			1	
Advanced age (≥35)	878	95(10.80%%)	6 (0.68%)	0.49 (0.39–0.62)	***	0.17	1.18 (0.85–1.65)	0.313
Parity		***	* (0.037)					
Primiparous (Ref)	1275	408 (32.00%)	3 (0.23%)	1			1	
Multiparous	1722	107 (6.20%)	14 (0.81%)	0.14 (0.11–0.18)	***	−1.79	0.17 (0.13–0.22)	***
History of abortion		0.825	-				-	-
No (Ref)	2541	435 (17.10%)	13 (0.51%)	1				
Yes	456	80 (17.50%)	4 (0.87%)	1.03 (0.79–1.34)	0.825	−0.08	-	-
Mode of conception		***	-					
Spontaneous (Ref)	2654	505 (19.01%)	15 (0.56%)	1				
ART	343	10 (2.92%)	2 (0.58%)	0.13 (0.07–0.24)	***	1.08	2.94 (0.76–11.43)	0.119
Mode of birth		***	-			-	-	-
SVB (Ref)	2184	0 (0.00%)	16 (0.73%) *	1				
Operative birth (OB)	813	515 (63.35%)	1 (0.12%)	nie---	0.980	-	-	-
Rupture of membrane		***	* 0.046			-	-	-
SROM (Ref)	1281	0 (0.00%)	10 (0.78%)	1				
tPROM	892	0 (0.00%)	1 (0.11%)	1.00 (0.001–99)	1.000	-	-	-
AROM	824	515 (62.50%)	6 (0.72%)	nie---	0.985	-	-	-

Variable(s) entered in step 1: Maternal age, Parity, History of abortion, Mode of conception 2, Intervention, BIG2 Mortality, Place of birth, and pooled outcome measures (Intervention × mortality). R^2^-Nagelkerke R-square, β- Regression coefficient, (-) indicates not fit for the model, (nie---) indicates not computed for the model, i.e., not indicated for enumeration (nie). Abbreviations and acronyms: AROM—Artificial rupture of membrane, ART—Assisted reproductive technologies, OB—Operative birth, SGA—Small for gestational age, Ref—reference group, SROM—spontaneous rupture of membrane (at term), SVB—Spontaneous vaginal birth, and tPROM—At term pre-labour rupture of membrane. Variables at *p* < 0.20 fixed value threshold on binary logistic were fitted to the multivariable logistic regression model. *Model 1*: Crude odds ratio. *Model 2*: Adjusted for maternal and neonatal factors. * *p* < 0.05; *** *p* < 0.001.

## Data Availability

Data that support the findings of this study (Tauffer database) are available; however, certain restrictions apply regarding the availability of these data, which were used under license for the current study, and are, therefore, not publicly available.

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
