# Peer review of "Characteristics of Homebirth in Hungary: A Retrospective Cohort Study"

_ijerph, 2022, doi:10.3390/ijerph191610461_

Round 1

Reviewer 1 Report

Thank you for letting me review this interesting paper on homebirth in Hungary, a very controversial subject in that country.

I have some comments for the authors:

General:
The paper could benefit from correction by a native English speaker.

Introduction:
In the first few lines, the authors refers to "cases" and "females" , instead of "women". Why not just use women?
I personally feel that a discussion of homebirth in Hungary would need to include the conviction and sanctions against Agnes Gereb in 2012. Also, some more clarification of the legal situation in Hungary is necessary, since Google tells us: 
In Hungary, it is legal for women to give birth at home. But any medical professional who helps those women — such as midwives — can be criminally charged.

Materials and methods:
It would be good to stress that the control group was matched for characteristics that make women eligible for homebirth (the "low risk group"). This is not completely clear at present. And why are both groups not the same size? It wasn't a case-control match then. I would imagine that with large numbers of hospital births it should be feasible to find a match for each case?
It would also help to be told more about the caregivers. Are they always midwives at home? Do these midwives work solo or as a group? Only at home or also in hospitals? How is their working relationship with obstetricians?
Why did you not use NICU admission as an outcome variable? Also, to me it is unclear what is meant by determinants. Some of these factors like parity and age are givens and therefore not outcomes. Also, how about maternal outcomes like PPH? Simply put, elements of starting points and endpoints seem mixed up, and there seem to be no maternal endpoints like PPH or perineal damage, and perinatal outcome seems to mostly be mortality, which should be very rare in any setting.
I do not understand the section on case-mix. I agree that low Apgar score is an outcome and should be excluded. But the other 3 are high risk factors (if known) and should exclude women from homebirth. So they should not occur in this study population?

Results:
Lines 168-172 seem to contradict each other. Were women with spontaneous conception more likely to give birth at home or not?
I am rather surprised by the high percentage of Apgar scores >9 at 5 minutes. Are Hungarian midwives perhaps a little optimistic? Also, this doesn’t seem to be a clinically significant difference.
In Table 2 I am very surprised by almost 40% LGA. How can this be? That is 4x normal, and this is a low risk population even! But the average birthweight is a very normal 3500 grams…
And 20% above the 99th percentile? How is that even mathematically possible?
Also, the percentages of most complications seem incredibly low, like perineal lacerations. Those are much higher in literature. Or are there many episiotomies? Those do not seem to be listed.
Table 3: what are non-obstetric complications? And why are there so many? I am now realizing as I read this, that it is unclear to me when women are transferred to a hospital. Which reasons are used? Meconium? Failure to progress? Pain relief?
Lines 220-221: why would someone have a homebirth if they intended to have an operative birth?
You state that in homebirth the intervention rate was 0.11%. That is hard to believe. What do you classify as an intervention? Because for instance 1.4% had a third stage hemorrhage. Were no interventions used for that? And what about the 11% transfers? Are those not interventions?
For a reader unfamiliar with the system in Hungary this is very difficult to follow.
Figure 3 and Table 4 or impossible for me to understand and should be left out or changed to make them easier to read.

Discussion:
In Lines 270-272 you again state that older mothers are more likely to have a home birth, but I wonder if this is corrected for parity?
I also don’t understand Line 272-274. Which other study?
Lines 272-277 are perhaps true, but I don’t see how they fit in this narrative.
I also don’t understand how you can say that maternal complications are more prevalent in homebirths. I see the opposite in your Tables.
I am not sure I agree that a transfer rate is an indication of an integrated system. Can you explain?
Line 310-311: in The Netherlands the current homebirth rate is actually historically low. It has never been this low before.
The argument for strength does not seem to be a real strength.
Limitations: can you name a few variables that might have been missed?

Conclusion:
The conclusion starts with a strange sentence. Why should it be on health care professionals to increase population growth?

Reviewer 2 Report

Thank you for giving me to review your manuscript. This manuscript is interesting and scientifically meaningful for considering the impact of homebirth on health outcomes in Hungary. Regarding the contents, the following revision should be considered.

The title should be more clearly described by removing “comparative.”

The expression of % and rate should be revised based on the proper analysis in the abstract and main document.

In the introduction, the term homebirth should be defined clearly as a keyword in this article.

The second paragraph of the introduction should contain an explanation of the outcome of pregnancy clearly.

The fourth paragraph of the introduction should delineate the concrete situations of home birth outside of Hungary to embark on the research question.

In the introduction, effective interprofessional collaboration should be driven for a safe home birth. The authors should describe interprofessional collaboration among healthcare professionals from developing and developed countries. The authors should describe international conditions of interprofessional collaboration, referring to the subsequent studies.

- Reeves, S., A. Xyrichis, and M. Zwarenstein, Teamwork, collaboration, coordination, and networking: Why we need to distinguish between different types of interprofessional practice. J Interprof Care, 2018. 32(1): p. 1-3.

- Ohta, R., Y. Ryu, and M. Yoshimura, Realist evaluation of interprofessional education in primary care through transprofessional role play: what primary care professionals learn together. Education for Primary Care, 2020: p. 1-9.

- Vedam, S., et al., Transfer from planned home birth to hospital: improving interprofessional collaboration. J Midwifery Womens Health, 2014. 59(6): p. 624-34.

- Saxell, L., S. Harris, and L. Elarar, The Collaboration for Maternal and Newborn Health: interprofessional maternity care education for medical, midwifery, and nursing students. J Midwifery Womens Health, 2009. 54(4): p. 314-20.

In the introduction, the researchers should show the research question clearly.

The method section should clearly describe one primary outcome and some secondary outcomes.

The sample calculation should be described in the analysis part.

In the discussion, the first paragraph should contain a summary of the results.

The discussion should describe the research findings in international contexts. As the same as the background, this is a critical point for the publication of international journals.

Round 2

Reviewer 2 Report

The manuscript has been considerably improved. I think that this paper is suited for inclusion in our journal.